

# Robustness of autoencoders for establishing psychometric properties based on small sample sizes: results from a Monte Carlo simulation study and a sports fan curiosity study

Yen-Kuang Lin[1], Chen-Yin Lee[2] and Chen-Yueh Chen[3]

[1] Graduate Institute of Athletics and Coaching Science, National Taiwan Sport University, Taoyuan, Taiwan
[2] Department of Applied Foreign Languages, MingDao University, Changhua, Taiwan
[3] Department of Leisure and Recreation Industry Management, National Taiwan Sport University, Taoyuan, Taiwan

## ABSTRACT

**Background**. The principal component analysis (PCA) is known as a multivariate statistical model for reducing dimensions into a representation of principal components. Thus, the PCA is commonly adopted for establishing psychometric properties, *i.e.,* the construct validity. Autoencoder is a neural network model, which has also been shown to perform well in dimensionality reduction. Although there are several ways the PCA and autoencoders could be compared for their differences, most of the recent literature focused on differences in image reconstruction, which are often sufficient for training data. In the current study, we looked at details of each autoencoder classifier and how they may provide neural network superiority that can better generalize non-normally distributed small datasets.

**Methodology**. A Monte Carlo simulation was conducted, varying the levels of non-normality, sample sizes, and levels of communality. The performances of autoencoders and a PCA were compared using the mean square error, mean absolute value, and Euclidian distance. The feasibility of autoencoders with small sample sizes was examined.

**Conclusions**. With extreme flexibility in decoding representation using linear and non-linear mapping, this study demonstrated that the autoencoder can robustly reduce dimensions, and hence was effective in building the construct validity with a sample size as small as 100. The autoencoders could obtain a smaller mean square error and small Euclidian distance between original dataset and predictions for a small non-normal dataset. Hence, when behavioral scientists attempt to explore the construct validity of a newly designed questionnaire, an autoencoder could also be considered an alternative to a PCA.

Corresponding authors
Chen-Yin Lee, inin1210@gmail.com
Chen-Yueh Chen, chenchenyueh@ntsu.edu.tw

# INTRODUCTION

Selecting a proper sample size is critical when planning an empirical study. The minimum sample size is often calculated based on selected statistical procedures. For example, inferential statistics based on an independent-sample $t$-test can apply the formula, $N \geq \left(\frac{1.96}{\delta}\right)^2 * \sigma^2$, to achieve the minimum sample size to detect a difference with a satisfactory probability. However, the feasibility of using an autoencoder to establish psychometric properties is not as straightforward. Although an autoencoder has not been used as often to evaluate psychometric properties compared to its alternative, the principal component analysis (PCA), autoencoders have been used in a vast array of scientific fields for dimensional reduction.

## Dimensional reduction

Dimensional reduction refers to the representation of high-dimensional information into a lower-dimensional space without losing an appreciable amount of data. Dimensional reduction can be applied to many methods of multivariate data analyses. For example, factor analysis (FA) is a tool to model interrelationships among items. In an FA, the focus is to partition the variance into either common variance or unique variance. By factor extraction, an FA can reduce the number of variables explaining the variance–covariance among items. Thus, much of the information in the original data can be retained but with fewer dimensions. As a result, this statistical tool, like so many other multivariate models, plays a superlative role in the fields of education, manufacturing industry, bioinformatics, and computer science. Two major challenges are encountered in dimension-reduction tasks. First, in order to evaluate the effectiveness of the dimensional reduction, a method of data visualization is often conducted to visualize the information in the data (*Young, Valero-Mora & Friendly, 2011*). However, data can only be visualized by humans in very limited lower dimensions of 2D or 3D. Thus, it will sometimes be impossible to graphically evaluate results of dimensional reduction. Second, although the dimension of a dataset can effectively be reduced by a stepwise algorithm, including a forward selection algorithm, backward elimination, stepwise procedure, *etc.*, the impact of each variable being examined is based on its contribution in improving the model fit. However, these stepwise algorithms are only valid when all of the dimensions are independent of each other (*Wang, 2008*). As a result, alternatives, like an FA, partial least squares, and PCA, can also be adopted. In contrast to a stepwise algorithm where nuisance variables are excluded, the FA and PCA retain all of the original items and combine them to form latent factors or principal components.

In contrast to the PCA and similar algorithms, the autoencoder is in essence a three-layer neural network, which was developed in the 1990s and which has been studied by many researchers. The association between auto-association and singular value decomposition was first examined by *Bourlard & Kamp (1988)*. *Kramer (1991)* attempted to conduct a nonlinear PCA using auto-associative neural networks. Most studies related to autoencoders were restricted to one or two hidden layers mainly due to training difficulties (*Wang et al., 2014*; *Wang, He & Prokhorov, 2012*). *Le (2013)* built a nine-layer sparse autoencoder to show that the network could be sensitive to higher-level concepts. Despite recent developments

with different architectures of autoencoders, creating a good reconstruction without losing significant information remains a challenging task due to high-dimensionality, small sample sizes, non-linearity, and complex data types.

## Construct validity

Construct validity is a psychological property defined as the degree to which a measure assesses the theoretical construct intended to be measured (*Cronbach & Meehl, 1955*). One cannot assess confounding influences of random error without estimating the construct validity when designing a questionnaire. Evaluation of what defines a psychological construct is the determinant of the test performance. A better representation of the latent psychological construct is desirable for almost all psychological tests. Thus, construct validity, in general, is considered the most fundamental aspect of psychometrics. *Campbell & Fiske (1959)* proposed two views of construct validity: convergent validity and discriminant validity. Convergent validity refers to the degree of agreement among measurements of the same constructs that should be related based on theory. Discriminant validity refers to the distinction of concepts that are not supposed to be related and are in fact, unrelated. Campbell and Fiske developed four steps based on inspecting the multitrait-multimethod (MTMM) matrix to operationally define convergent validity and discriminant validity. Since the concept of construct validity was introduced, an extensive effort has been made ever since to seek numerical representations of the construct validity. Starting with Douglas Jackson who employed a component analysis as an integral part of the development of psychological measures, the PCA has become a standard method for questionnaire development (*Jackson, 1970*). Traditionally, the PCA and FA are two of the most often employed statistical procedures in the social behavioral sciences commonly used to suggest factor profiles as latent constructs (*Sherman, 1986*; *Yoon et al., 2019*; *Fontes et al., 2017*).

In FAs, the confirmatory factor analysis (CFA) and exploratory factor analysis (EFA) are two methods that facilitate the transition from many observed variables to a smaller number of latent variables. Both FA models are commonly used to address the construct validity. The CFA is a tool that researchers can adopt to test the validity by comparing alternatively proposed a priori models at the latent factor level. Advantages of the CFA for being more informative than Campbell & Fiske's criteria are that it provides a statistical justification of the model fit and the degree of fit for the convergent validity and divergent validity (*Bagozzi, Yi & Phillips, 1991*).

In addition to the CFA, there are several statistical models based upon which the factors explored by a questionnaire are validated. For example, the PCA is a mathematical algorithm in which observations are described by several inter-correlated quantitatively dependent variables. A PCA is the default data-reduction technique in SPSS software and was adopted by many researchers, including *Mohammadbeigi, Mohammadsalehi & Aligol (2015)*, and *Parker, Bindl & Strauss (2010)*. A PCA can be conducted to examine the construct validity due to the PCA's ability to integrate the full bivariate cross-correlation matrix of all item-wise measurements through dimension reduction. Its goal is to extract important information from the total number of observed variables, and represent it as a

set of new orthogonal variables called principal components. These principal components, or latent variables, summarize the observed data table and display the pattern of similarity of the observations (*Abdi & Williams, 2010*).

Most educational researchers, behavioral scientists, and social science researchers treat uncorrelated principal components as independent identities. However, the property of principal components being independent of each other only holds when the principal components are uncorrelated, and the multivariate items are normally distributed (*Kim & Kim, 2012*). If the input data are not normally distributed, the variance explained by one of the traits will overlap that of another trait. PCAs have also been criticized due to limited linear mapping representation.

Based on the underlying definition of dimensional reduction, it is less informative to aggregate all scales into a single latent variable score. A set of items or scales may share similar conceptual underpinnings but not necessarily be identical (*Stangor, 2014*). Using a PCA, a large number of items can be reduced to fewer components with possibly more variance explained than with other methods of factoring (*Hamzah, Othman & Hassan, 2016*).

## Autoencoder

Autoencoders are characterized by their function of extracting important information and representing it in another space. Such a network consists of three symmetrical layers: input, hidden, and output layers (*Hinton & Zemel, 1994*). An autoencoder attempts to approximate the original data so that the output is similar to the input after feed-forward propagation. The input is projected to the hidden layer that is commonly designed to be of a lower dimensionality. After information is passed through the hidden layer, the output of the network should ideally resemble the original input as closely as possible. As a result, the latent space contains all of the necessary information to describe the data (*Ladjal, Newson & Pham, 2019*). In a simple autoencoder framework, each neuron is fully connected to all neurons in the previous layer, where neurons in a single layer function completely independently and share no connections. As illustrated by from *Meng, Ding & Xue (2017)*, autoencoder tries to learn function $S(\cdot)$ such that:

$$S_{W,W',b1,b2}(X) \approx X \tag{1}$$

$W$ is weight matrix connected input layer and hidden layer while $W'$ is weight matrix connected hidden layer and output layer. $b_1$ and $b_2$ are bias vectors of hidden layer and output layer. $S(\cdot)$ can be divided into two phases: from input layer to hidden layer is encoding phase Eq. (2) and from hidden to output layer is decoding phase Eq. (3).

$$h = f(W \times X + b_1) \tag{2}$$

$$Y = g(W' \times h + b2) \tag{3}$$

## Autoencoder *versus* PCA

When one is interested in establishing the construct validity, it is often intuitive to apply a PCA to extract latent factors. It quickly becomes apparent that the PCA shares lots of similarities with an autoencoder. Both methods can serve as tools for feature generation and selection by their ability to reduce dimensions. Despite autoencoder neural networks bearing a significant resemblance to PCAs, there is one major difference between these two networks. In contrast to a PCA, an autoencoder applies a non-linear transformation to the input, and so the autoencoder could be more flexible. That is, although the PCA can effectively reduce the linear dimensionality, it still suffers when relationships among the variables are not linear. Aside from the linearity restriction, a PCA may fail by its loose assumption about the input data distribution. As *Shlens (2014)* pointed out, even though the PCA algorithm is in essence completely nonparametric, because a PCA is unconcerned with the source of the data, it might not capture key features of data variations. That is, a PCA makes no assumptions about the distribution of the data. However, only when the data are assumed to be normal from a multivariate perspective will the joint distribution of the principal components be normal from a multivariate perspective. Then, the principal components will have an obvious geometrical interpretation where the first component can be determined by locating the chord of maximum distance in the ellipsoid $(x - \mu)^T \Sigma^{-1} (x - \mu) = $ constant (*Chatfield & Collins, 1981*).

As a result, we can directly compare various forms of autoencoders to a PCA when we attempt to build the construct validity of a small sample when the data are not normally distributed. Four different forms of autoencoders are considered in this study, including a simple autoencoder with a single-layered autoencoder and three other candidates briefly described as follows.

## Tie-weighted autoencoder

An autoencoder is a neural network with a symmetrical structure. Although the input is compressed and the output is reconstructed through its latent-space representation, there is no guarantee that the weights of the encoder and decoder are identical. Thus, we can impose an additional optimization restriction so that the weights of the decoder layer are tied to the weights of the encoder layer. By tying the weights, the number of parameters that needs to be trained and the risk of overfitting are reduced. Tie-weight autoencoder is the one we set it to be $W = W'$ from Eqs. (2) and (3) (*Meng, Ding & Xue, 2017*).

## Deep autoencoder

In the autoencoder framework, there is no limitation on the number of layers for the encoder or decoder. That is, the autoencoder can go deep and can be implemented with a stack of layers. Theoretically, the more hidden layers there are, the more features can be learned from the hidden layers. Although the layers can be stacked, the layers are often designed to remain symmetrical with respect to the central layers.

## Independent encoded autoencoder

A preferable feature of a PCA is that the weight vectors are independent of each other. If orthogonality is imposed, each encoded feature explains unique information, and a

smaller number of encoder layers can be achieved. Thus, we also adopted an orthogonal autoencoder (COAE) for comparison, which is capable of simultaneously extracting latent embedding and predicting the clustering assignment (*Wang et al., 2019*).

### Sample size

There are only a few studies concerning the requirement of the sample size on the dimensional reduction performance of a PCA. So there is no consensus as to how large is large enough for conducting a PCA. *Forcino Frank (2012)* found that a too-small sample size is more likely to lead to erroneous conclusions. *Manjarrés-Martínez et al. (2012)* tested the stability performance of three ordination methods in terms of their bootstrap-generated sampling variances. Bootstrap resampling techniques are used to generate larger samples which may provide more-precise evaluations of the sampling error. Some researchers recommend sample sizes in relation to the number of variables or correlation structure. For example, *Hatcher & O'Rourke (2013)* suggested that the sample size should be larger than five times the number of variables. *Hutcheson & Sofroniou (1999)* recommended that a minimum of $n = 150$ is required for a high community correlation structure. *Mundfrom, Shaw & Tian (2005)* found that $n > 100$ was required for medium community while *MacCallum et al. (2001)* achieved satisfactory results even for data with numbers of items greater than the sample size. In contrast, *Yeung & Ruzzo (2001)* showed that a PCA is not suitable for dimensionality reduction tasks when $p$ is greater than $n$. The performance of a PCA is worsened when a nonlinear relationship is present with limited samples.

A deep neural network is competitive in solving nonlinear dimensional reductions for high-dimensional data. Although it may seem legitimate that a massive amount of data is required to train a deep neural network, some researchers claim that deep learning can still be adopted even if $n$ is small. *Seyfioğlu & Gürbüz (2017)* compared a convolutional autoencoder and two convolutional neural networks, VGGNet and GoogleNet, in terms of the training sample sizes. They found that when the sample size exceeded 650, the convolutional autoencoder outperformed transfer learning and random initialization.

Although there is a great number of dimensionality reduction algorithms being developed, the feasibility of their use with small-sample, non-normal data is still unknown. A limited sample size and an non-linear data distribution may also increase the likelihood of overfitting and decrease the accuracy. To overcome the pitfalls of the sample size issue, the principal objective of the current study was to examine the influence of sample size on the latent structure of PCAs and autoencoders using a Monte Carlo simulation. The performances of the PCA and various transformations by autoencoders were evaluated using both simulated data and a real dataset pertaining to quantifying the concept of curiosity.

## MATERIALS & METHODS

The detailed design of the simulation consisted of three major states (Fig. 1): data generation, dimensionality-reduction algorithms, and performance evaluation. The input dataset was first divided into two sub-datasets: a training set and testing set. Then various forms of autoencoders along with a PCA were applied to select desirable encoded

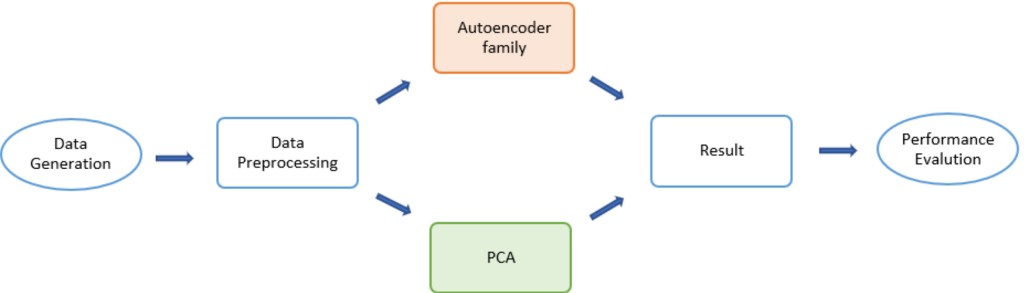

**Figure 1** Flow chart for the Monte Carlo (MCMC) simulations.

dimensions of attributes. Finally, for latent dimensionality classification of the obtained reduced-dimensional data, a reconstruction error was applied to evaluate the algorithms.

## Data generation

A Monte Carlo simulation was used for this study. To avoid the overfitting issue, separate datasets were used for the dimension-reduction algorithms. Each simulated dataset was divided into two parts: 80% of samples were used as a training set, and the remaining 20% were used as a test set. The dimension-reduction techniques, including various type of autoencoders and a PCA, were trained on the training dataset. After the classifier was built, the testing data were used to test the effectiveness of the classifier. Each simulated dataset was simulated based on its degree of non-normality, correlation among items, and sample size. Correlation matrices of continuous variables, each representing a questionnaire correlation structure, were generated for each condition by three manipulated variables: the degree of communality, the degree of non-normality, and the sample size. The data generation python code was stored in Github, and can be assessed at https://github.com/robbinlin/data-generation-/blob/e94206b6a16751961c3db57fbe93017dc050d746/data_generation_20211005.ipynb.

## Non-normality

A variety of mathematical algorithms have been developed over the years to simulate conditions of non-normal distributions (*Fan et al., 2002*; *Fleishman, 1978*; *Ramberg et al., 1979*; *Schmeiser & Deutsch, 1977*). *Fleishman (1978)* also introduced a method for generating sample data from a population with desired degrees of skewness and kurtosis. That method uses a cubic transformation to transform a standard univariate normally distributed variable to obtain a nonnormal variable with specified degrees of skewness and kurtosis. The transformation developed by Fleishman takes the form of

$$Y = a + bZ + cZ^2 + dZ^3;$$ (4)

where Y is the transformed non-normal variable, Z is a standard normal random variable, and a, b, c, and d are coefficients needed for transforming the unit around the unit normal to a non-normal variable with specified degrees of population skewness and kurtosis (*Byrd, 2008*).

These coefficients were tabulated in *Fleishman (1978)* for selected combinations of degrees of skewness and kurtosis. *Fleishman (1978)* derived a system of nonlinear equations that given the target distribution mean, variance, skewness, and kurtosis, could be solved for coefficients to produce a third-order polynomial approximation to the desired distribution (*Fan et al., 2002*).

## Correlation structure

In order to generate non-normal correlated observation, the interaction between inter-variable correlation and degree of non-normality needs to be considered since difference combination of inter-variable correlations and non-normality conditions would cause sample data to deviate from the specified correlation pattern. The abovementioned Fleishman's method can be extended to multivariate non-normal data with a specified correlation (*Wicklin, 2013*). For example, two non-normal variables, Y1 and Y2, can be generated with specified skewness and kurtosis from Eq. (4) *i.e.*,

$$Y_1 = a_1 + b_1 Z + c_1 Z^2 + d_1 Z^3 \qquad (5)$$

$$Y_2 = a_2 + b_2 Z + c_2 Z^2 + d_2 Z^3 \qquad (6)$$

Coefficients $a_1$, $b_1$, $c_1$, $d_1$, $a_2$, $b_2$, $c_2$, and $d_2$ can be derived from Fleishman's table once the degree of skewness and kurtosis are known. After these coefficients ($a_i$, $b_i$, $c_i$, $d_i$) are found, the intermediate correlations could be derived by specifying $R_{x1x2}$, the population correlation between two non-normal variables $Y_1$ and $Y_2$. *Vale & Maurelli (1983)* demonstrated that through the following relationship,

$$R_{x_1 x_2} = \rho (b_1 b_2 + 3 b_1 d_2 + 3 d_1 b_2 + 9 d_1 d_2) + \rho^2 (2 c_1 c_2) + \rho^3 (6 d_1 d_2) \qquad (7)$$

the intermediate correlation, $\rho$, can be derived. These coefficients in the Fleishman power transformation above are required to derive intermediate correlations, $\rho$. After all of the intermediate correlation coefficients are assembled into an intermediate correlation matrix, this intermediate correlation matrix is then used to extract factor patterns to transform uncorrelated items into correlated items (*Vale & Maurelli, 1983*). *Kaiser & Dickman (1962)* presented a matrix decomposition procedure that imposes a specified correlation matrix on a set of uncorrelated random normal variables specified with population correlations *R* as represented by the imposed correlation matrix. The basic matrix decomposition procedure takes the following form (*Kaiser & Dickman, 1962*):

$$\hat{R}_{k*N} = F_{k*k} \times \hat{X}_{k*N} \qquad (8)$$

where k is the number of variables, N is the number of sample, $\hat{R}$ is the resulting data matrix, F is the principal component factor pattern coefficients obtained by principal component factorization to the desired population matrix *R*, and X are uncorrelated k random variables, with N observations. After the intermediate correlations are derived from Vale and Maurelli 's formula using iterative Newton–Raphson method, we then apply

the assembled intermediate correlation matrix to Kaiser and Dickman's method as R in order to generate N sample.

In the current study, three correlation structures for 15 items were simulated to approximate three communalities of high, in which communalities were assigned values of 0.8; wide, in which they could have value from 0.6 to 0.9; and low, in which they could have value from 0.3 and 0.5. The communality estimate is the estimated proportion of variance in each variable that is accounted for *Mamdoohi et al. (2016)*. These estimates reflect the proportion of variation in that variable explained by the latent factors (*Yong & Pearce, 2013*). In other words, a high communality means that if we perform multiple regression of curiosity against the three common factors, we obtain a satisfactory proportion of the variation in curiosity explained by the factor model. These estimates reflect the variance of a variable in common with all others together. The communality is denoted by $h^2$ and is the summation of the squared correlations of the variable with the factors (*Barton, Cattell & Curran, 1973*). The formula for deriving the communalities is $h_j^2 = a_{j1}^2 + a_{j2}^2 + \ldots a_{jm}^2$ where a denotes the loadings for j variables. The communality levels correspond inversely to levels of importance of unique factors. High communalities imply several variables load highly on the same factor and the model error is low (*MacCallum et al., 2001*). These three correlation structures were designed to mimic a three-factor solution and are presented as follows.

## Correlation matrix for high communality

$$
\begin{bmatrix}
1 & 0.7 & 0.7 & 0.7 & 0.7 & 0.2 & 0.2 & 0.2 & 0.2 & 0.2 & 0.2 & 0.2 & 0.2 & 0.2 & 0.2 \\
0.7 & 1 & 0.7 & 0.7 & 07 & 0.2 & 0.2 & 0.2 & 0.2 & 0.2 & 0.2 & 0.2 & 0.2 & 0.2 & 0.2 \\
0.7 & 0.7 & 1 & 0.7 & 0.7 & 0.2 & 0.2 & 0.2 & 0.2 & 0.2 & 0.2 & 0.2 & 0.2 & 0.2 & 0.2 \\
0.7 & 0.7 & 0.7 & 1 & 0.7 & 0.2 & 0.2 & 0.2 & 0.2 & 0.2 & 0.2 & 0.2 & 0.2 & 0.2 & 0.2 \\
0.7 & 0.7 & 0.7 & 0.7 & 1 & 0.2 & 0.2 & 0.2 & 0.2 & 0.2 & 0.2 & 0.2 & 0.2 & 0.2 & 0.2 \\
0.2 & 0.2 & 0.2 & 0.2 & 0.2 & 1 & 0.7 & 0.7 & 0.7 & 0.7 & 0.2 & 0.2 & 0.2 & 0.2 & 0.2 \\
0.2 & 0.2 & 0.2 & 0.2 & 0.2 & 0.7 & 1 & 0.7 & 0.7 & 0.7 & 0.2 & 0.2 & 0.2 & 0.2 & 0.2 \\
0.2 & 0.2 & 0.2 & 0.2 & 0.2 & 07 & 0.7 & 1 & 0.7 & 0.7 & 0.2 & 0.2 & 0.2 & 0.2 & 0.2 \\
0.2 & 0.2 & 0.2 & 0.2 & 0.2 & 07 & 0.7 & 0.7 & 1 & 0.7 & 0.2 & 0.2 & 0.2 & 0.2 & 0.2 \\
0.2 & 0.2 & 0.2 & 0.2 & 0.2 & 0.7 & 0.7 & 0.7 & 0.7 & 1 & 0.2 & 0.2 & 0.2 & 0.2 & 0.2 \\
0.2 & 0.2 & 0.2 & 0.2 & 0.2 & 0.2 & 0.2 & 0.2 & 0.2 & 0.2 & 1 & 0.7 & 0.7 & 0.7 & 0.7 \\
0.2 & 0.2 & 0.2 & 0.2 & 0.2 & 0.2 & 0.2 & 0.2 & 0.2 & 0.2 & 0.7 & 1 & 0.7 & 0.7 & 0.7 \\
0.2 & 0.2 & 0.2 & 0.2 & 0.2 & 0.2 & 0.2 & 0.2 & 0.2 & 0.2 & 0.7 & 0.7 & 1 & 0.7 & 0.7 \\
0.2 & 0.2 & 0.2 & 0.2 & 0.2 & 0.2 & 0.2 & 0.2 & 0.2 & 0.2 & 0.7 & 0.7 & 0.7 & 1 & 0.7 \\
0.2 & 0.2 & 0.2 & 0.2 & 0.2 & 0.2 & 0.2 & 0.2 & 0.2 & 0.2 & 0.7 & 0.7 & 0.7 & 0.7 & 1
\end{bmatrix}
$$

### Correlation matrix for intermediate communality

$$
\begin{bmatrix}
1 & 0.9 & 0.8 & 0.7 & 0.6 & 0.1 & 0.1 & 0.1 & 0.1 & 0.1 & 0.1 & 0.1 & 0.1 & 0.1 & 0.1 \\
0.9 & 1 & 0.7 & 0.6 & 0.5 & 0.1 & 0.1 & 0.1 & 0.1 & 0.1 & 0.1 & 0.1 & 0.1 & 0.1 & 0.1 \\
0.8 & 0.7 & 1 & 0.5 & 0.4 & 0.1 & 0.1 & 0.1 & 0.1 & 0.1 & 0.1 & 0.1 & 0.1 & 0.1 & 0.1 \\
0.7 & 0.6 & 0.5 & 1 & 0.3 & 0.1 & 0.1 & 0.1 & 0.1 & 0.1 & 0.1 & 0.1 & 0.1 & 0.1 & 0.1 \\
0.6 & 0.5 & 0.4 & 0.3 & 1 & 0.1 & 0.1 & 0.1 & 0.1 & 0.1 & 0.1 & 0.1 & 0.1 & 0.1 & 0.1 \\
0.1 & 0.1 & 0.1 & 0.1 & 0.1 & 1 & 0.9 & 0.8 & 0.7 & 0.6 & 0.1 & 0.1 & 0.1 & 0.1 & 0.1 \\
0.1 & 0.1 & 0.1 & 0.1 & 0.1 & 0.9 & 1 & 0.7 & 0.6 & 0.5 & 0.1 & 0.1 & 0.1 & 0.1 & 0.1 \\
0.1 & 0.1 & 0.1 & 0.1 & 0.1 & 0.8 & 0.7 & 1 & 0.5 & 0.4 & 0.1 & 0.1 & 0.1 & 0.1 & 0.1 \\
0.1 & 0.1 & 0.1 & 0.1 & 0.1 & 0.7 & 0.6 & 0.5 & 1 & 0.3 & 0.1 & 0.1 & 0.1 & 0.1 & 0.1 \\
0.1 & 0.1 & 0.1 & 0.1 & 0.1 & 0.6 & 0.5 & 0.4 & 0.3 & 1 & 0.1 & 0.1 & 0.1 & 0.1 & 0.1 \\
0.1 & 0.1 & 0.1 & 0.1 & 0.1 & 0.1 & 0.1 & 0.1 & 0.1 & 0.1 & 1 & 0.9 & 0.8 & 0.7 & 0.6 \\
0.1 & 0.1 & 0.1 & 0.1 & 0.1 & 0.1 & 0.1 & 0.1 & 0.1 & 0.1 & 0.9 & 1 & 0.7 & 0.6 & 0.5 \\
0.1 & 0.1 & 0.1 & 0.1 & 0.1 & 0.1 & 0.1 & 0.1 & 0.1 & 0.1 & 0.8 & 0.7 & 1 & 0.5 & 0.4 \\
0.1 & 0.1 & 0.1 & 0.1 & 0.1 & 0.1 & 0.1 & 0.1 & 0.1 & 0.1 & 0.7 & 0.6 & 0.5 & 1 & 0.3 \\
0.1 & 0.1 & 0.1 & 0.1 & 0.1 & 0.1 & 0.1 & 0.1 & 0.1 & 0.1 & 0.6 & 0.5 & 0.4 & 0.3 & 1
\end{bmatrix}
$$

### Correlation matrix for weak communality

$$
\begin{bmatrix}
1 & 0.5 & 0.5 & 0.5 & 0.5 & 0.3 & 0.3 & 0.3 & 0.3 & 0.3 & 0.3 & 0.3 & 0.3 & 0.3 & 0.3 \\
0.5 & 1 & 0.5 & 0.5 & 0.5 & 0.3 & 0.3 & 0.3 & 0.3 & 0.3 & 0.3 & 0.3 & 0.3 & 0.3 & 0.3 \\
0.5 & 0.5 & 1 & 0.5 & 0.4 & 0.3 & 0.3 & 0.3 & 0.3 & 0.3 & 0.3 & 0.3 & 0.3 & 0.3 & 0.3 \\
0.5 & 0.5 & 0.5 & 1 & 0.3 & 0.3 & 0.3 & 0.3 & 0.3 & 0.3 & 0.3 & 0.3 & 0.3 & 0.3 & 0.3 \\
0.5 & 0.5 & 0.4 & 0.3 & 1 & 0.3 & 0.3 & 0.3 & 0.3 & 0.3 & 0.3 & 0.3 & 0.3 & 0.3 & 0.3 \\
0.3 & 0.3 & 0.3 & 0.3 & 0.3 & 1 & 0.5 & 0.5 & 0.5 & 0.5 & 0.3 & 0.3 & 0.3 & 0.3 & 0.3 \\
0.3 & 0.3 & 0.3 & 0.3 & 0.3 & 0.5 & 1 & 0.5 & 0.5 & 0.5 & 0.3 & 0.3 & 0.3 & 0.3 & 0.3 \\
0.3 & 0.3 & 0.3 & 0.3 & 0.3 & 0.5 & 0.5 & 1 & 0.5 & 0.4 & 0.3 & 0.3 & 0.3 & 0.3 & 0.3 \\
0.3 & 0.3 & 0.3 & 0.3 & 0.3 & 0.5 & 0.5 & 0.5 & 1 & 0.3 & 0.3 & 0.3 & 0.3 & 0.3 & 0.3 \\
0.3 & 0.3 & 0.3 & 0.3 & 0.3 & 0.5 & 0.5 & 0.4 & 0.3 & 1 & 0.3 & 0.3 & 0.3 & 0.3 & 0.3 \\
0.3 & 0.3 & 0.3 & 0.3 & 0.3 & 0.3 & 0.3 & 0.3 & 0.3 & 0.3 & 1 & 0.5 & 0.5 & 0.5 & 0.5 \\
0.3 & 0.3 & 0.3 & 0.3 & 0.3 & 0.3 & 0.3 & 0.3 & 0.3 & 0.3 & 0.5 & 1 & 0.5 & 0.5 & 0.5 \\
0.3 & 0.3 & 0.3 & 0.3 & 0.3 & 0.3 & 0.3 & 0.3 & 0.3 & 0.3 & 0.5 & 0.5 & 1 & 0.5 & 0.4 \\
0.3 & 0.3 & 0.3 & 0.3 & 0.3 & 0.3 & 0.3 & 0.3 & 0.3 & 0.3 & 0.5 & 0.5 & 0.5 & 1 & 0.3 \\
0.3 & 0.3 & 0.3 & 0.3 & 0.3 & 0.3 & 0.3 & 0.3 & 0.3 & 0.3 & 0.5 & 0.5 & 0.4 & 0.3 & 1
\end{bmatrix}
$$

Factor patterns were estimated with one of these three correlation matrices. These pattern matrices were then adopted to generate 15 correlated normal variables with specified population correlation coefficients, variable means, standard deviations (SDs), skewness, and kurtosis.

### Sample size

The sample size was chosen based on the recommendations of *Mundfrom, Shaw & Tian (2005)*. Sample size increments were according to the following algorithm:

- When $n < 200$, it was increased by 10;
- When $n < 500$, it was increased by 50;

## Dimensionality-reduction algorithms

- Simple autoencoder: A simple autoencoder is an autoencoder with two main components: The Encoder and the decoder in addition to the latent-space representation layer also known as the bottle neck layer. Linear activation function and SGD optimizer were adopted. The encoding and decoding algorithms will be chosen to be parametric functions and to be differentiable with respect to the loss function (*Chollet, 2016*). By minimizing the reconstruction loss, the parameters of the encoder and decoder can be optimized. The basic autoencoder, which refers to simple autoencoder in section 1.4 is a autoencoder with a single fully-connected neural layer as encoder and as decoder (Fig. 2A).
- Tie-weighted autoencoder: A tie-weighted autoencoder is a single-layer autoencoder with three neurons in addition to a decoder and an encoder layer. A restriction is imposed so that the weights of the encoder and decoder are identical.
- Deep autoencoder: In order to demonstrate that the autoencoder algorithms do not have to limit to a single layer as encoder or decoder, we could instead use a stack of layers, so called "deep autoencoder" in 'Autoencoder versus PCA' A deep autoencoder is a sevenlayer autoencoder. The first layer is a dense layer with 11 neurons follow by two layers each with six neurons. Before the decoder, a bottleneck layer with three neurons is included. The architecture is depicted in Fig. 2B.
- Independent encoded autoencoder: With this custom layer, we impose penalty on the sum of off-diagonal elements of the encoded features covariance to create uncorrelated features as well as applying orthogonality on both encoder and decoder Weights.

## Real dataset

In this subsection, we implemented the PCA and autoencoders with a real dataset—a Sports Fan Curiosity dataset–to see how the autoencoders differed from the PCA and provide visualization results. More specifically, we compared the ability to build psychometric properties between the autoencoders and the PCA on a Sports Fan Curiosity questionnaire. Curiosity is considered a fundamental intrinsic motivational subdomain for initiating human exploratory behaviors in many field of study, such as psychology, education, and sports. The Sports Fan Curiosity questionnaire is commonly used in the field of sports management, which is a subset of questionnaires for measuring behavioral intentions. This subcategory was designed to measure and quantify the construct of curiosity. Although leisure management has received greater attention, little is known about how the curiosity construct can be realized with a structured questionnaire. We deployed the autoencoders for this questionnaire and evaluated their ability to identify the latent construct. Items of the Sports Fan Curiosity questionnaire are listed in Table 1. The data set along with the python code was stored in Google Colab and Google drive, and can be assessed at https://colab.research.google.com/drive/1pC8A10sRVUHDttkLF2ATb51CcpnIzD6u?usp=sharing.

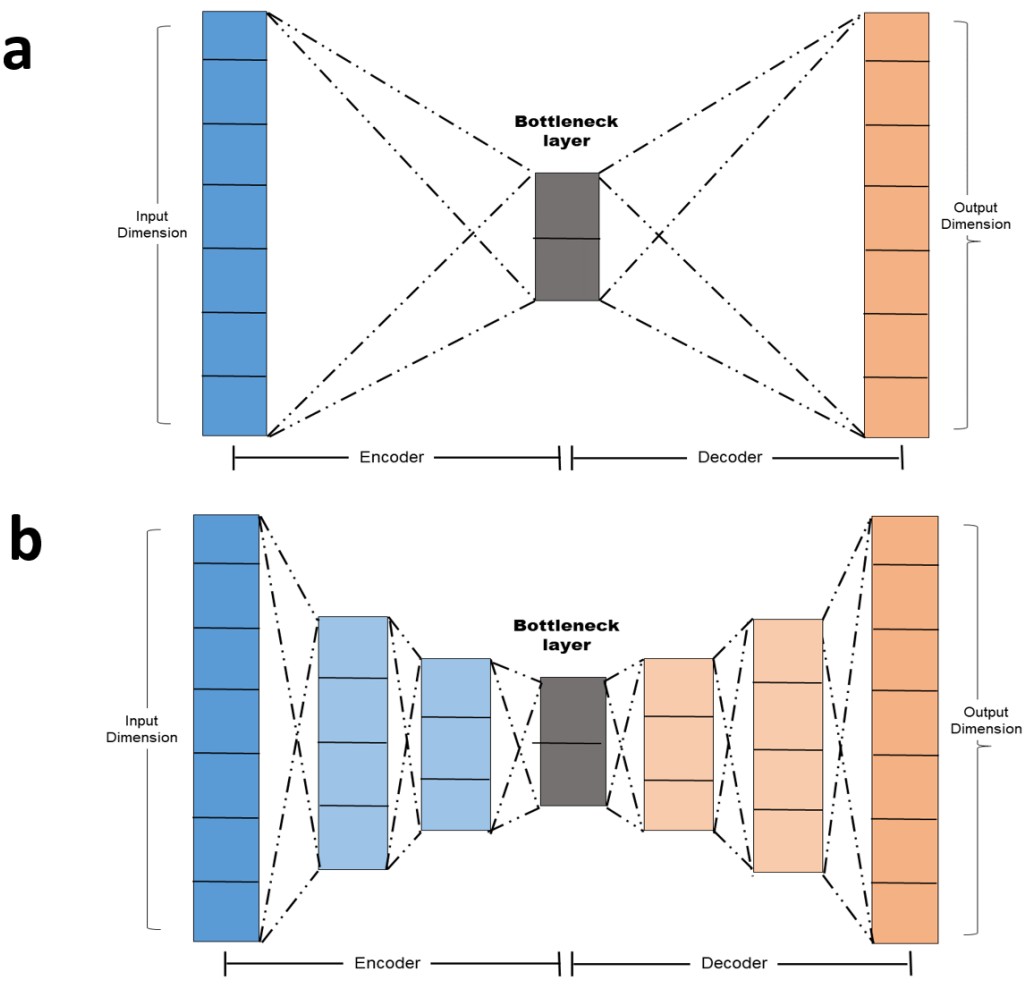

**Figure 2  (A–B) Autoencoder architecture.**

## Performance metrics
### *Reconstruction error*

The performance of the establishment of construct validity was evaluated using the mean square error (MSE), mean absolute error (MAE), and the average of the Euclidean distance.

Denote inputs to the network by $\underline{X}$ and the outputs of the network by $\underline{y}$. Then the network can be described by mapping from inputs to outputs: $\underline{y} = f(x)$. Reconstruction of the input is then a mapping from outputs to inputs $\hat{X} = g(y)$. It is then reasonable to measure the reconstruction error as $\underline{X} - \hat{X}$ with a given error function, $\varepsilon(x)$. In this study, the reconstruction error is defined as the average of the squares of the errors for subject n with dimension m, *i.e.*,

$$\text{MSE} = \frac{1}{\text{n}} \cdot \frac{1}{m} \sum_{i=1}^{n} \sum_{j=1}^{m} \left( X_{i,j} - \hat{x}_{i,j} \right)^2$$

**Table 1  Items on the curiosity questionnaire.**

| Item | Description |
|---|---|
| 1 | I enjoy collecting and calculating statistics for my favorite basketball team. |
| 2 | I often imagine how my favorite basketball team is playing to defeat their opponent. |
| 3 | I enjoy exploring my favorite basketball stadiums or facilities. |
| 4 | Watching basketball games with my friends is joyful. |
| 5 | I enjoy reading articles about basketball players, teams, events, and games. |
| 6 | I am interested in learning how much it costs to build a brand-new basketball stadium. |
| 7 | When I miss a game, I often look for information on television, the internet, or newspaper to catch the game results. |
| 8 | I am interested in learning how large a basketball court is. |
| 9 | I enjoy probing deeply into basketball. |
| 10 | I am eager to learn more about basketball. |
| 11 | I enjoy any movement that occurs during a basketball game. |

### *Mean absolute error (MAE)*

The MAE is defined as the deviation between the paired predicted value $(\widehat{X}_i)$ and original value (X). That is, it is the average of the absolute errors for subject n with dimension m:

$$\mathrm{MAE} = \frac{\sum_{i=1}^{n}\sum_{j=1}^{m}\left|\widehat{x_{i,j}} - x_{i,j}\right|}{nm}$$

### *Normalized-euclidean distances (NEDs)*

The average of the Euclidean distance between each pair of samples in X and $\widehat{X}_i$ was also calculated to accommodate possible missing values for subject n with dimension m.

$$\mathrm{NED} = \sqrt{\sum_{i=1}^{n}\sum_{j=1}^{m}\left(\frac{\hat{x}_{i,j}}{\left|\hat{x}_{i,j}\right|} - \frac{x_{i,j}}{\left|x_{i,j}\right|}\right)}$$

In summary, a $3 \times 3 \times 27$ factorial design was implemented according to the manipulated variables of community level, non-normality level, and sample size, resulting in a total of 243 population conditions (Table 2). Each of the scenario was simulated for 10 times.

## RESULTS

The efficacy of the autoencoders was identified using a Monte Carlo simulation that manipulated five population parameters. The correlation structure was determined with three levels of communality (high, wide, and low), three levels of normality (normal, slightly un-normal, and un-normal), and nine levels of hidden layers of neurons. One should be

**Table 2   Parameters for generating non-normal data.**

| Scenario no. | Mean | Standard deviation | Skewness | Kurtosis |
|---|---|---|---|---|
| 1 | 0 | 1 | 0 | 0 |
| 2 | 0 | 1 | 1 | 3 |
| 3 | 0 | 1 | 2 | 20 |

**Table 3   Performance metrics for three communality conditions.**

| Metric | Algorithm | High communality | | Wide communality | | Low communality | |
|---|---|---|---|---|---|---|---|
| | | Mean | SD | Mean | SD | Mean | SD |
| MSE | Simple encoder | 0.343 | 0.175 | 0.382 | 0.203 | 0.458 | 0.146 |
| | Tied encoder | 0.034 | 0.009 | 0.034 | 0.010 | 0.034 | 0.009 |
| | PCA | 0.230 | 0.160 | 0.254 | 0.196 | 0.337 | 0.148 |
| | Deep autoencoder | 0.032 | 0.019 | 0.036 | 0.022 | 0.044 | 0.017 |
| | Independent autoencoder | 0.036 | 0.021 | 0.040 | 0.023 | 0.048 | 0.018 |
| MAE | Simple encoder | 0.444 | 0.112 | 0.460 | 0.128 | 0.523 | 0.086 |
| | Tied encoder | 0.126 | 0.018 | 0.126 | 0.018 | 0.127 | 0.017 |
| | PCA | 0.357 | 0.116 | 0.356 | 0.144 | 0.443 | 0.101 |
| | Deep autoencoder | 0.107 | 0.030 | 0.110 | 0.034 | 0.128 | 0.024 |
| | Independent autoencoder | 0.113 | 0.031 | 0.117 | 0.035 | 0.133 | 0.025 |
| NED | Simple encoder | 2.288 | 0.166 | 2.277 | 0.179 | 2.226 | 0.144 |
| | Tied encoder | 1.053 | 0.076 | 1.055 | 0.074 | 1.060 | 0.073 |
| | PCA | 4.730 | 0.303 | 4.745 | 0.341 | 4.634 | 0.291 |
| | Deep autoencoder | 1.146 | 0.086 | 1.144 | 0.094 | 1.114 | 0.079 |
| | Independent autoencoder | 1.138 | 0.099 | 1.132 | 0.104 | 1.111 | 0.085 |

**Notes.**

MSE, mean squared error; PCA, principal component analysis; MAE, mean absolute error; NED, non-Euclidian distance; SD, standard deviation.

aware that the current study focused on the feasibility of analyzing small samples. As a result, sample sizes beyond 1,000 were not considered.

Overall, the autoencoders had smaller reconstruction errors compared to the PCA counterpart. Specifically, the tied-weight autoencoder, deep autoencoder, and independent-feature autoencoder outperformed the PCA in all three communality conditions. The Tied autoencoder was the most stable algorithm among all candidates, as its SD for reconstruction was the smallest (Table 3). In general, the autoencoder family produced smaller MAEs and NEDs compared to the PCA, except for the simple encoder. The simple autoencoder generated larger MSEs and MAEs for all three communality conditions. When the communality was low, performances of the deep autoencoder, tied-weight autoencoder, and independent autoencoder were less affected, in contrast to the PCA and simple encoder.

With respect to the input data distributions, it was observed that the performances of the autoencoder family and PCA were not affected by non-normality conditions for this small sample size simulation. However, it was interesting to observe that for

**Table 4  Performance metrics for three normality conditions.**

| Metric | Algorithm | Normal | | Slightly un-normal | | Un-normal | |
|---|---|---|---|---|---|---|---|
| | | Mean | SD | Mean | SD | Mean | SD |
| MSE | Simple encoder | 0.397 | 0.182 | 0.394 | 0.181 | 0.392 | 0.185 |
| | Tied encoder | 0.034 | 0.009 | 0.034 | 0.009 | 0.034 | 0.010 |
| | PCA | 0.277 | 0.174 | 0.275 | 0.175 | 0.269 | 0.177 |
| | Deep autoencoder | 0.038 | 0.020 | 0.038 | 0.020 | 0.037 | 0.019 |
| | Independent autoencoder | 0.042 | 0.021 | 0.041 | 0.021 | 0.041 | 0.021 |
| MAE | Simple encoder | 0.484 | 0.117 | 0.473 | 0.114 | 0.471 | 0.115 |
| | Tied encoder | 0.128 | 0.018 | 0.126 | 0.018 | 0.125 | 0.017 |
| | PCA | 0.393 | 0.130 | 0.382 | 0.128 | 0.380 | 0.127 |
| | Deep autoencoder | 0.117 | 0.032 | 0.114 | 0.031 | 0.114 | 0.030 |
| | Independent autoencoder | 0.123 | 0.033 | 0.121 | 0.032 | 0.120 | 0.031 |
| NED | Simple encoder | 2.305 | 0.158 | 2.263 | 0.164 | 2.223 | 0.166 |
| | Tied encoder | 1.075 | 0.073 | 1.053 | 0.074 | 1.041 | 0.072 |
| | PCA | 4.787 | 0.299 | 4.699 | 0.314 | 4.622 | 0.314 |
| | Deep autoencoder | 1.155 | 0.086 | 1.134 | 0.087 | 1.116 | 0.087 |
| | Independent autoencoder | 1.147 | 0.095 | 1.126 | 0.096 | 1.108 | 0.096 |

**Notes.**

MSE, mean squared error; MAE, mean absolute error; NED, non-Euclidian distance; PCA, principal component analysis; SD, standard deviation.

data that was extremely un-normal, the tied-weight encoder, deep autoencoder, and independent-feature encoder outperformed the PCA in terms of the MSE and MAE. Of all autoencoder variations, the tied-weight encoder generated the smallest MSE (Table 4). We also evaluated reconstruction errors under different combinations of communality conditions and non-normality conditions. When input data were normally distributed, the resulting reconstruction errors were similar among the three communalities. However, if the correlation structure was low, the MSE increased for non-normal data, even if it only slightly deviated from a normal distribution (Fig. 3).

Regarding the sample size, the MSE for the autoencoder family decreased as the sample size increased. MSEs for the deep autoencoder, tied-weight encoder, and independent encoder were smaller than that for the PCA for a small sample size. The MSE appeared to decrease to a local minimum when the sample size reached 200 (Fig. 4). The simple encoder had the largest MSE, while the tied-weight encoder had the smallest MSE for all of the different sample sizes considered. On the other hand, the PCA had the largest Euclidean distance between the original data and its predictions, while the NEDs were smaller for the tied-weight encoder and deep autoencoder. We further analyzed reconstruction errors for all of the autoencoder family and PCA under different combinations of normality conditions and communality. MSEs for the tied-weight encoder continued to decrease until the sample size reached 200 regardless of the deviation from a normal distribution and communality conditions (Figs. 4 & 5). A similar trend was also found for the deep autoencoder. The MSE was negatively associated with the sample size, and the MSE stabilized for values of $n$ greater than 400. In contrast, the reconstruction error for the

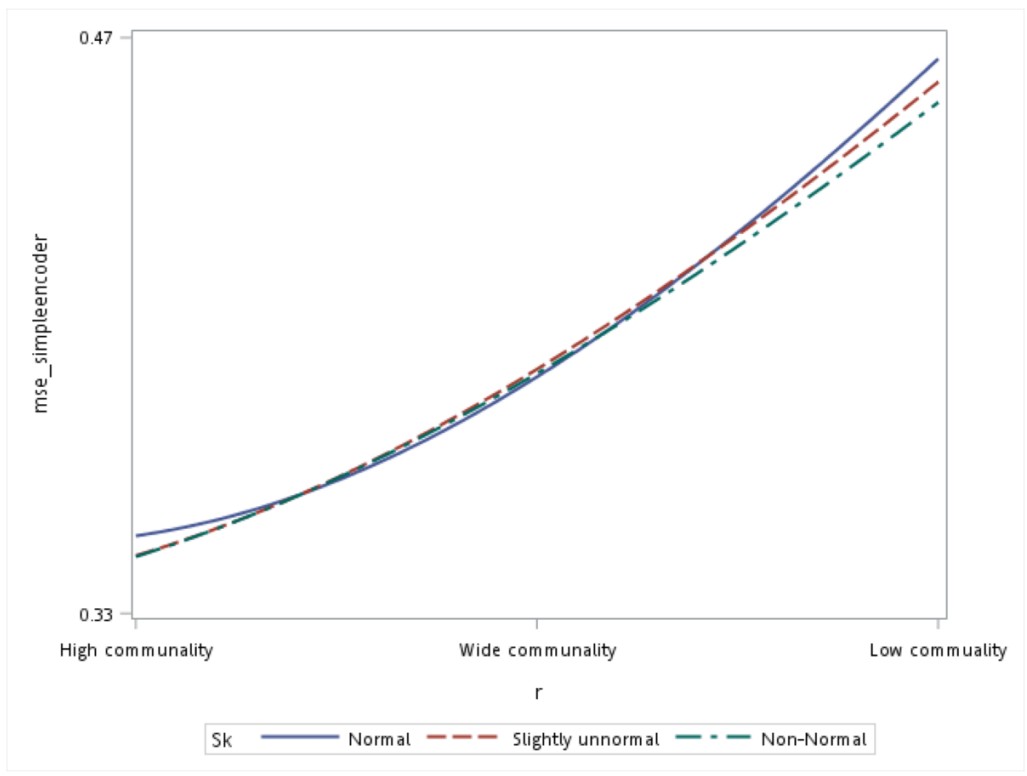

**Figure 3 Interaction of communality and non-normality condition on the mean squared error (MSE) for the simple autoencoder.**

PCA was much contaminated by the weak commonality condition. If the correlation structure was low communality, the MSE was always higher than those of the wide and high communality. A similar pattern was also observed for the MAE, as the PCA had a larger MAE when the correlation structure was low communality. In contrast, the deep autoencoder seemed to be insensitive to community conditions and deviations from normality in terms of the MAE compared to that obtained from PCA (Fig. 6).

There are several algorithms available for determining the number of retaining factors for a PCA, *e.g.*, a scree plot and parallel analysis. However, there are no guidelines to choose the size of the bottleneck layer in the autoencoder. From the simulation results, the autoencoder seemed to perform better compared to the PCA especially when k was small (Fig. 7). Even when the number of components was correctly specified by the PCA (which was $k = 3$ in our simulation), all of the autoencoder variations besides the simple encoder outperformed the PCA in terms of MSE.

## With a real dataset

We directly compared reconstruction errors of both the deep autoencoder and PCA on the Sports Fan Curiosity questionnaire. Based on previous simulation result, a random sample of 100 subjects was chosen from the original data of 400 subjects to examine the performance with small data. When the PCA was applied to the curiosity data, three

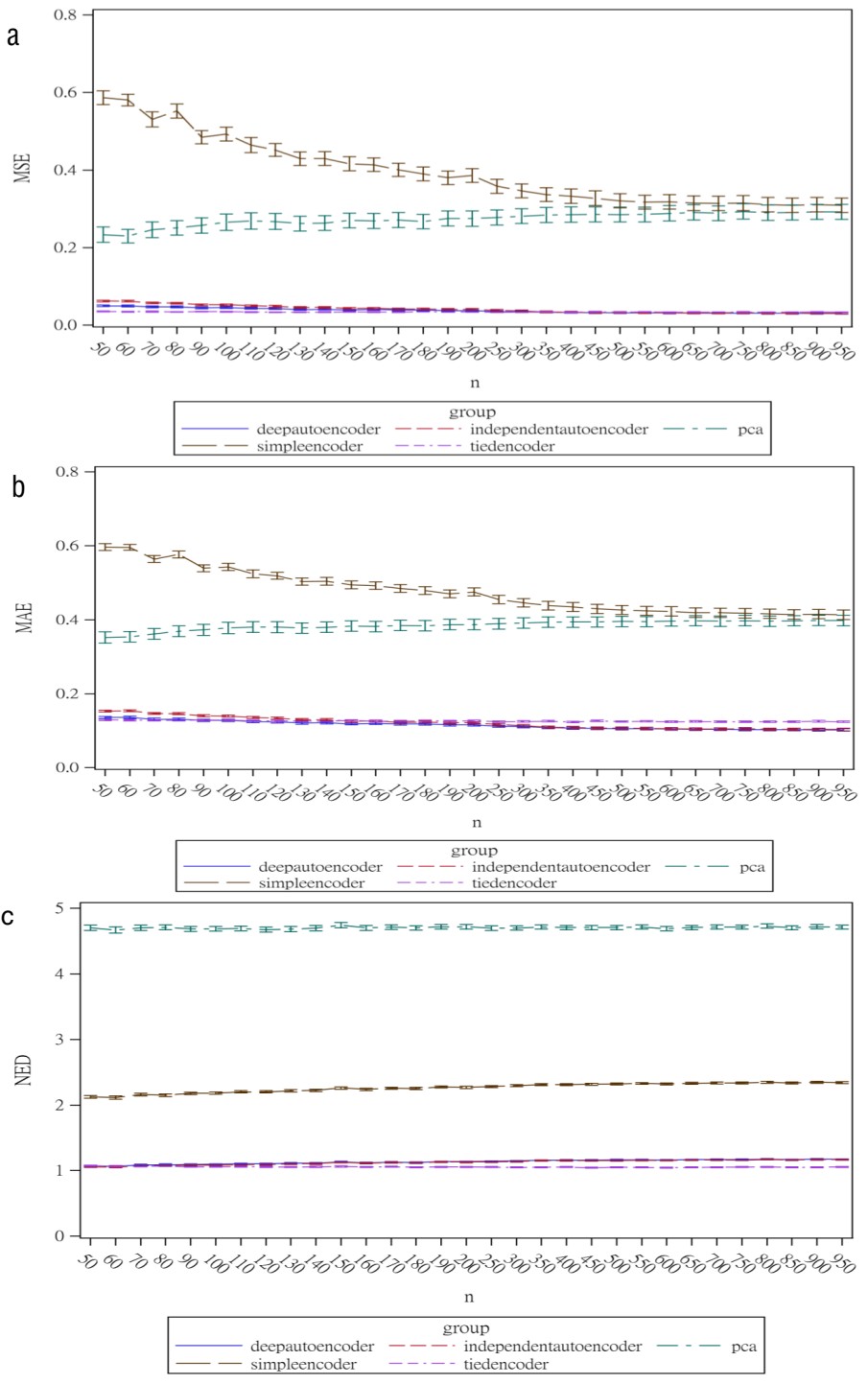

**Figure 4  Reconstruction error for the autoencoder family under different sample sizes.**

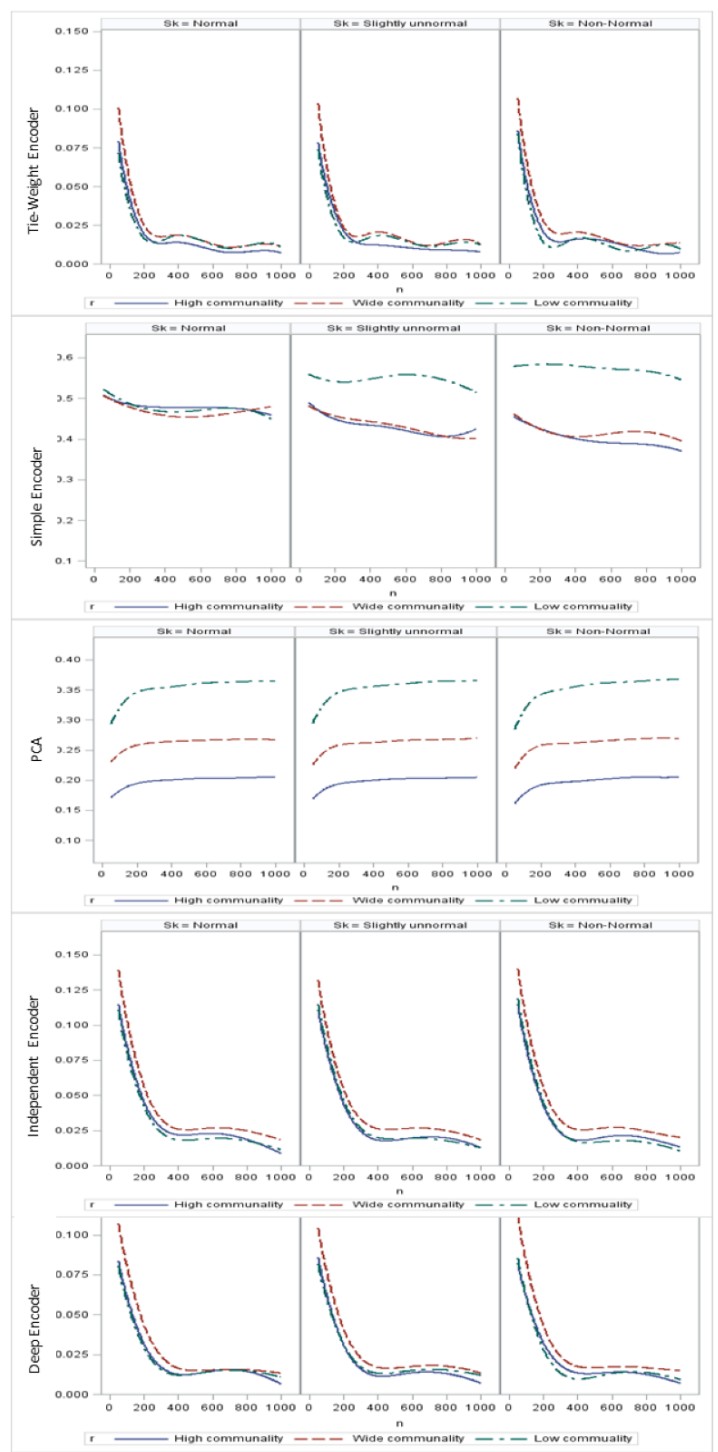

**Figure 5** **Mean square errors (MSEs) for different combinations of communality and normality.**

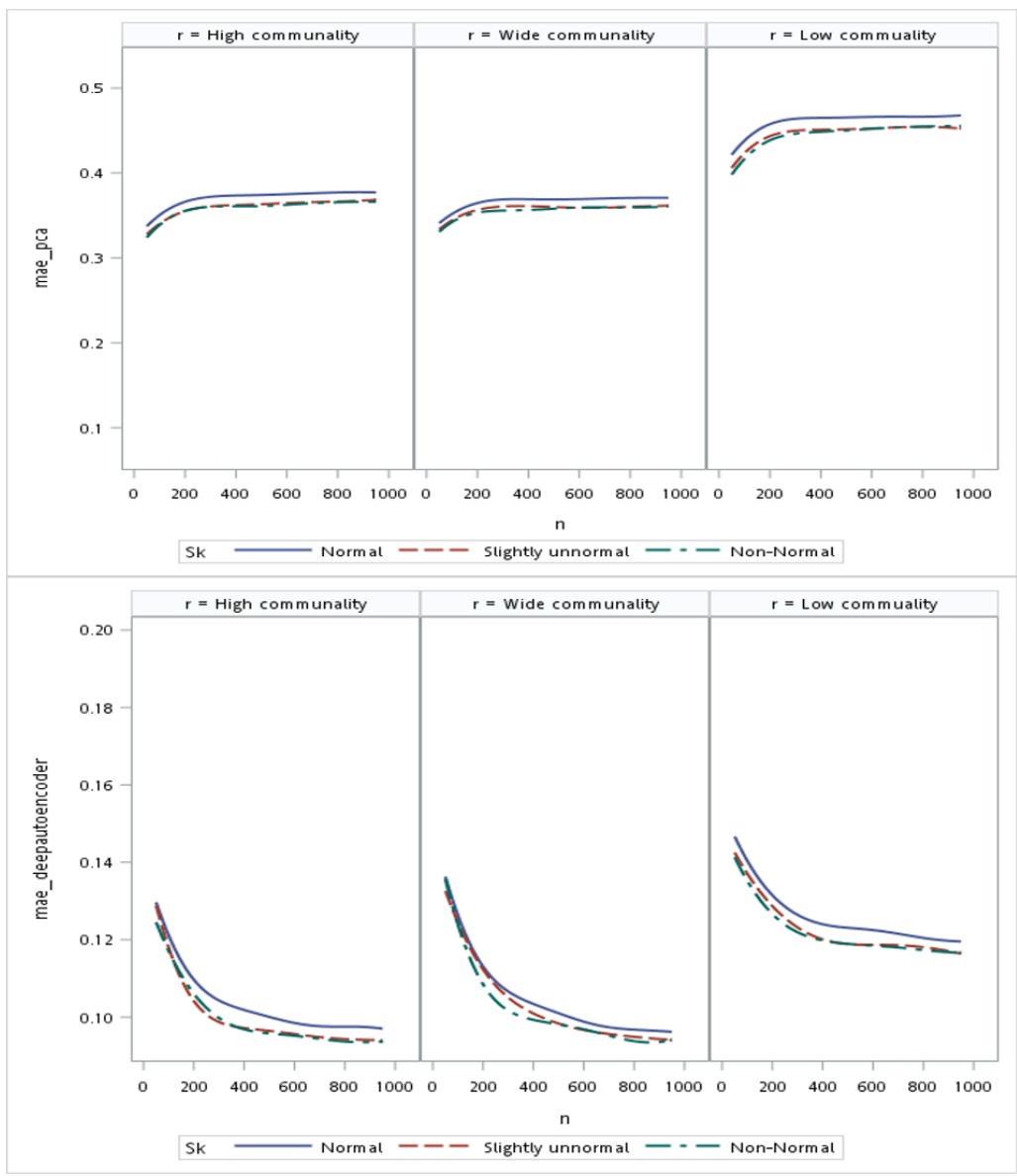

**Figure 6** Mean absolute errors (MAEs) for the principal component analysis (PCA) and deep autoencoder.

components were extracted with an $R^2$ of 0.53 and an MSE of 0.46 while $R^2$ reached to 60.1% and the MSE decreased to 0.36 for the autoencoder. We adopted $t$-distributed stochastic neighboring entities (t-SNEs) to visualize the reduced dimension. It was observed that the three latent components could separate the clusters using the PCA, but there was clearly some information that overlapped the extracted components. In contrast to the

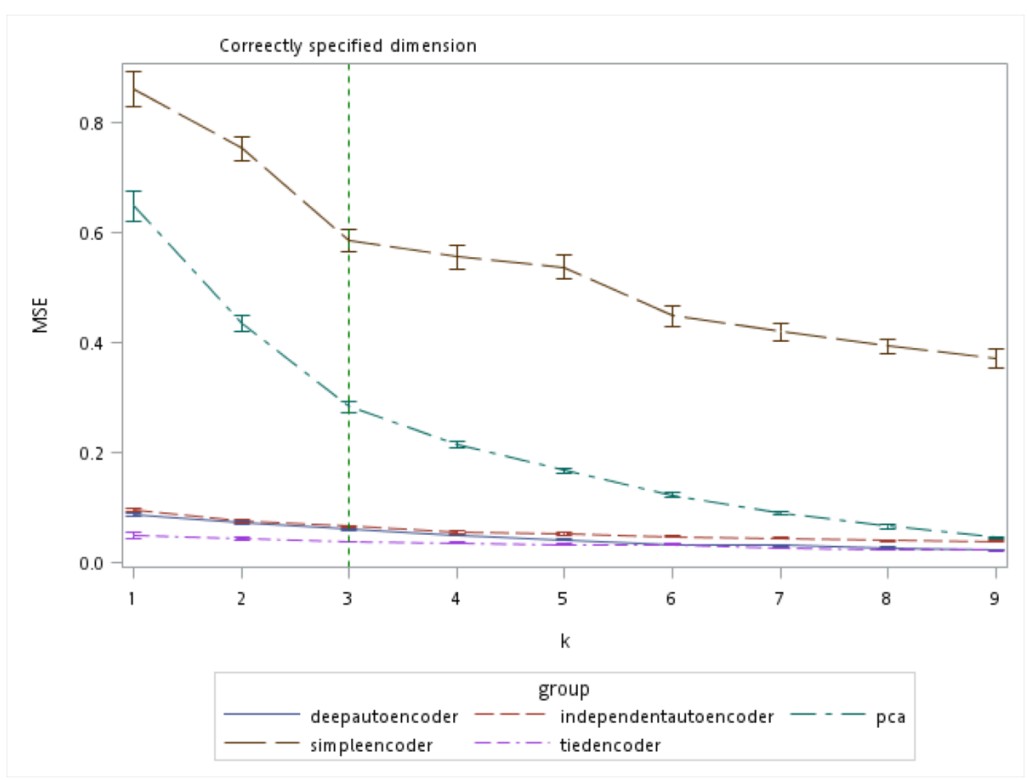

**Figure 7** Mean squared errors (MSEs) of the autoencoder and principal component analysis (PCA) under various extracted components.

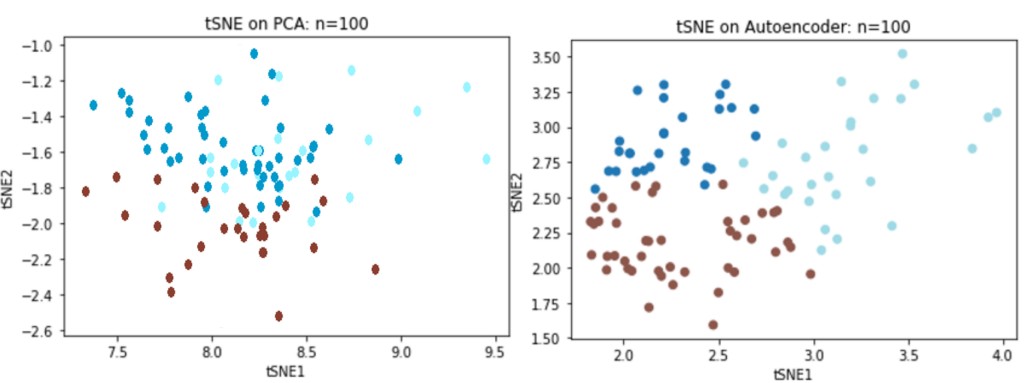

**Figure 8** TSNE visualizations for PCA and Autoencoder.

PCA, the autoencoder could better separate the three underlying constructs, as we saw that there was a significant improvement over the PCA (Fig. 8).

Regarding the sport fan curiosity questionnaire, if the questionnaire is construct valid, all items together well represent the underlying construct. Based on the weights estimated from the autoencoder's bottleneck layers, each item could be classified into one of the three

latent dimensions by its weights in absolute values. For example, the item "I enjoy collecting and calculating statistics of my favorite basketball team" has the highest correlation with construct 1. Similarly, we see that item "Watching basketball games with my friends is joyful" has the highest correlation with construct 2. As such, autoencoders can elucidate how different items and constructs relate to one another and help develop new theories. For example, in Table 5, the items "I enjoy collecting and calculating statistics of my favorite basketball team," "I enjoy reading articles about basketball players, teams, events, and games," "I am eager to learn more about basketball" and "I enjoy any movement that occurs during a basketball game" appear to have large coefficient on one latent construct, which we assign as "Learning Motivation" factor. "Watching basketball games with my friends is joyful," "I enjoy probing deeply into basketball," and "I often imagine how my favorite basketball team is playing to defeat their opponent" depict the "Social Interaction "factor. The three extracted constructs were knowledge, social interaction, and facility. From the weighted estimate from the bottleneck layer in the autoencoder, each item was classified into one of three latent constructs based on the largest weights among the three clusters. As a result, the first component consisted of items 1, 5, 10, and 11. Items 2, 3, and 6 were classified into cluster 2, while items 4, 7, 8, and 9 belonged to cluster 3. Based on the items in each cluster, three latent constructs were identified, *i.e.,* learning motivation, social interaction, and facility (Table 5).

The results from the current study were conceptually equivalent to the three constructs of sport fan curiosity scale developed by *Park, Ha & Mahony (2014)*, *i.e.,* specific information, general information and sport facility information. The slight differences between findings from the study and the work of Park et al. may result from different research contexts. More specifically, the real dataset used in the study was collected from a specific sports context (*i.e.,* basketball) whereas Park et al. developed the sport fan curiosity scale in a general sport context. However, the findings from the study with small sample size ($n = 100$) yielded psychometrically similar factor structure to the work of Park et al. with a much larger sample size ($n = 407$). Consequently, the effectiveness and efficiency of the proposed methodology in the study enrich the relevant literature theoretically and practically.

## DISCUSSION

The PCA and autoencoders are two common ways of reducing the dimensionality of a high-dimensional feature space. The PCA is a linear-transformation algorithm which projects features onto an orthogonal basis and thus, generates uncorrelated features. An autoencoder is an unsupervised learning technique that can be adopted to tackle the task of representative learning. That is, a bottleneck layer in the network is imposed so that knowledge is compressed through the bottleneck layer, and the output is a representation of the original input. A correlation structure that exists in the data can be learned and consequently leveraged when forcing the input through the network's bottleneck. This characteristic of an autoencoder allows the encoder to compress information into a low-dimensional space to model complicated non-linear associations.

One of the major objectives of the current study was to provide a feasibility analysis for adopting a neural network model for establishing construct validity as a psychometric

**Table 5   Absolute values of bottleneck weights on the Curiosity dataset.**

**a. Bottleneck weights estimates from the population.**

| Learning motivation | Social interaction | Facility | |
|---|---|---|---|
| **0.329** | 0.029 | 0.237 | I enjoy collecting and calculating statistics of my favorite basketball team. |
| **0.474** | 0.178 | 0.293 | I enjoy reading articles about basketball players, teams, events, and games. |
| **0.335** | 0.158 | 0.261 | I am eager to learn more about basketball. |
| **0.68** | 0.003 | 0.227 | I enjoy any movement that occurs during a basketball game. |
| **0.385** | 0.433 | 0.415 | Watching basketball games with my friends is joyful. |
| **0.351** | 0.575 | 0.114 | I enjoy probing deeply into basketball. |
| **0.314** | 0.662 | 0.039 | I often imagine how my favorite basketball team is playing to defeat their opponent. |
| **0.125** | 0.519 | 0.469 | I enjoy exploring my favorite basketball stadiums or facilities. |
| **0.011** | 0.385 | 0.023 | I am interested in learning how much it costs to build a brand new basketball stadium. |
| **0.104** | 0.19 | 0.581 | I am interested in learning how large a basketball court is. |
| **0.099** | 0.127 | 0.231 | When I miss a game, I often look for information on television, the internet, or newspaper to catch the game results. |

**b. Bottleneck weights estimates from sample of $n = 100$.**

| Learning motivation | Social interaction | Facility | |
|---|---|---|---|
| **0.431** | 0.090 | 0.138 | I enjoy collecting and calculating statistics of my favorite basketball team. |
| **0.532** | 0.067 | 0.299 | I enjoy reading articles about basketball players, teams, events, and games. |
| **0.322** | 0.239 | 0.312 | I am eager to learn more about basketball. |
| **0.541** | 0.051 | 0.170 | I enjoy any movement that occurs during a basketball game. |
| **0.308** | 0.767 | 0.124 | Watching basketball games with my friends is joyful. |
| **0.089** | 0.440 | 0.100 | I enjoy probing deeply into basketball. |
| **0.355** | 0.685 | 0.077 | I often imagine how my favorite basketball team is playing to defeat their opponent. |
| **0.431** | 0.175 | 0.431 | I enjoy exploring my favorite basketball stadiums or facilities. |
| **0.117** | 0.203 | 0.615 | I am interested in learning how much it costs to build a brand new basketball stadium. |

**Table 5** (*continued*)

**b. Bottleneck weights estimates from sample of $n = 100$.**

| | | | |
|---|---|---|---|
| **0.126** | 0.344 | 0.383 | I am interested in learning how large a basketball court is. |
| **0.031** | 0.078 | 0.224 | When I miss a game, I often look for information on television, the internet, or newspaper to catch the game results. |

property. Our simulation results indicated that if a low communality correlation structure existed, the reconstruction error for the autoencoder would increase for the non-normal data scenario when the sample size was small. This result showed that although the computational resources for the neural network were generally more expensive, besides the simple encoder, the reconstruction error for the autoencoder was uniformly smaller across three different communality conditions as well as three different non-normality conditions.

## CONCLUSIONS

A common practical question in conducting factor analyses and PCAs is how many subjects are sufficient to obtain a reliable estimate. There are many rules of thumb proposed suggesting a certain absolute sample size, such as a minimum of 250 or 500. Some suggested a size-to-time ratio to be as high as 10 times as many subjects as variables. There is no universally applicable answer, but the answer depends upon the clarity of the structure being examined, *i.e.,* the communality of the variables. If there is well-structured communality among the variables, the size of the sample required goes down. As the number of items for a factor increase, the sample size needed decreases. That is, if the number of variables is high relative to the number of factors (*e.g.,*15:3), and the communality is high, then sample sizes as small as 60∼100 are adequate. In FA models, more subjects are always better, but what is more important is to have good markers for each factor (high communality) as well as many markers (a high item-to-factor ratio) than it is to increase the number of subjects. Unfortunately, although it will never be wrong advice to have many markers of high communality, if using an autoencoder to analyze the structure of items rather than tests, the communality requirement will tend to be low. In cases where communality is low and data are not linearly distributed, increasing the number of subjects is advised. Based on this simulation study, the autoencoder tended to perform relatively better when the neurons of the bottleneck, k, were small, which means that the same reconstruction accuracy of the original scale could be achieved with fewer components and hence a smaller dataset. This is important when dealing with many items or variables in a questionnaire. For all sample sizes being considered, the tied-weight encoder had relatively small reconstruction errors. Based on this Monte Carlo simulation, we demonstrated that it is feasible for an autoencoder to be used in psychometric research to establish the construct validity with a small sample size.

### Funding

The authors received no funding for this work.

### Competing Interests

The authors declare there are no competing interests.

### Author Contributions

- Yen-Kuang Lin conceived and designed the experiments, performed the experiments, analyzed the data, performed the computation work, prepared figures and/or tables, and approved the final draft.
- Chen-Yin Lee conceived and designed the experiments, prepared figures and/or tables, authored or reviewed drafts of the paper, and approved the final draft.
- Chen-Yueh Chen performed the experiments, prepared figures and/or tables, authored or reviewed drafts of the paper, and approved the final draft.

### Data Availability

The code is available at GitHub:

https://github.com/robbinlin/data-generation-/blob/7e1504e53385a4b3497a8d7d42137
858460bb019/Curiosity_example_(public_20211028).ipynb

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
