# Peer review of "Robustness of autoencoders for establishing psychometric properties based on small sample sizes: results from a Monte Carlo simulation study and a sports fan curiosity study"

_PeerJ Computer Science, doi:10.7717/peerj-cs.782_

## Round 0.1 · original submission · Major Revisions

As pointed out by the two referees, there are several unclear aspects in the present article. Both require a more rigorous scientific approach, with special care on the definitions and the mathematical aspects. Furthermore, other issues seem to show little attention to the final draft of the paper (missing references and wrong figure captions).

Please follow all the referee suggestions carefully and resubmit the paper.

Reviewer 1 ·

Basic reporting

The article under review tries to compare the performances of the PCA and autoencoders in the field of questionnaire analysis.

Although the work seems serious and well done
I have some reservations about the general form of the article that made its reading difficult.
Plus the exact definitions of the ento-encoders (see bellow).

Some definitions are not clear and a mathematical clear definition:
- What is communality and non-cummunality (see references problem bellow)
- The formula of line 325 (and in general all the metrics used) should be clarified as one wonders if the sum is running across the data-set or across the dimensions of the data.
- In section 2.4: what does the term "increased by" refer to? n is the number of samples, what is there to increase.

In general a clearer mathamtical presentation is needed.

I'm also concerned by the exact definitions of the autoencoders: In the colab

First many references are not listed in the reference section (for example Forcino (2012) Manjarres-Martinez et al. (2012) can not be found in the reference section). Some citation follow the form [number] and other follow the form "Author (Year)".

It is important to fix this citation problem.

Likewise, some figures are mis-referenced (Figure 8 in line 403 is inexistent and surely the authors meant figure 6 in line 390).

There is a bulletpoint in line 295 that has no text.

Experimental design

My main concern is about the definition of the auto-encoders: In the code submitted the activation are "linear" this means that the autoencoders boils down to an affine transform. How can an affine transform do better than the PCA in termes of MSE? I think this is because non linear auto-encoders overfit on the small number of samples used for training.

Judging by the code submitted, what is called deep-autoencoder has only one layer more than the autoencoder. This is not a really deep architecture. In the code submmited the deepautoencoder has two times 3-size layers while the text mesion one layer with dimension 6.

I also encourage the authors to disclose the data generation code (for synthetic data)

Validity of the findings

Pending the experimental clarification about the architectures I can not comment on the validity of the findings.

Reviewer 2 ·

Basic reporting

The paper compares PCA and Autoencoder as dimentionality reduction techniques, aiming at showing the superiority of autoencoders in case of small sample sizes (and departures from normality) in the context of psychometric survey data
As a preliminary observation, the title of the paper seems not appropriate since, its main content is a MC simulation study and only an application to psychometric data is presented. Moreover, this real data example is based on a single random sample draw from the original data set . The authors should either consider more real data sets or resampling several times form the single one considered averaging on the results also, the results on the whole data set (400obs) should be reported.
When applying the PCA, since the items are ordinal (the indication of the scale is missing…) I suggest to employ Polychoric correlations.

In general, the paper lacks definitions and details. While PCA is a very well known multivariate method , NN are in general considered black boxes , several structures are possible and none of the considered architecture are displayed in the paper, making very difficult to understand them and their differences.

Note that there’s no figure 8 in the paper and that citations need to be fixed as the follow different citation standards. Also, on. Line 295 there’s an extra bulletpoint

Experimental design

The three correlation structures are designed to mimic a 3 factor solution organizing 15 items into block of 5 showing the same correlation. For the high communality for example, the value of 0.8 and then always 0.1 for cross blocks items. I think that this design is too extreme and far from any real situation.
In section 2.4 (sample size) don’t understand what the sample size increments are meant for.

Validity of the findings

The paper is not clear enough to let me judge on the validity of the findings.

Architectures of the NN should be displayed and suggestions about the real data application followed to facilitate the reading and the evaluation

---

## Round 0.2 · accepted · Accept

Congratulations on the acceptance of your manuscript, and thanks for submitting your work to PeerJ Computer Science. I checked the requests the reviewers had made for the first version of the manuscript and the changes you made; it is my opinion that the work is worthy of publication.

Reviewer 2 ·

Basic reporting

The authors have satisfactorily responded to all my questions .
From my point of view, the paper is now suitable for publication.

Experimental design

The authors have modified the experimental design according to my remarks

Validity of the findings

In this new version the validity of the findings emerges in a clearer way.